# Wearable-Measured Sleep and Resting Heart Rate Variability as an Outcome of and Predictor for Subjective Stress Measures: A Multiple N-of-1 Observational Study

**DOI:** 10.3390/s23010332

**Published:** 2022-12-28

**Authors:** Herman J. de Vries, Helena J. M. Pennings, Cees P. van der Schans, Robbert Sanderman, Hilbrand K. E. Oldenhuis, Wim Kamphuis

**Affiliations:** 1Research Group Digital Transformation, Hanze University of Applied Sciences, 9747 AS Groningen, The Netherlands; 2Department of Human Behaviour & Training, Netherlands Organization for Applied Scientific Research (TNO), 3769 DE Soesterberg, The Netherlands; 3Department of Health Psychology, University Medical Center Groningen, 9700 AB Groningen, The Netherlands; 4Utrecht Center for Research and Development of Health Professions Education, University Medical Center Utrecht, 3584 CX Utrecht, The Netherlands; 5Department of Rehabilitation Medicine, University Medical Center Groningen, 9700 AB Groningen, The Netherlands; 6Research Group Healthy Ageing Allied Health Care and Nursing, Hanze University of Applied Sciences, 9747 AS Groningen, The Netherlands; 7Department of Psychology, Health and Technology, University of Twente, 7522 NB Enschede, The Netherlands

**Keywords:** wearables, heart rate variability, sleep, stress, time series analysis, police, ecological momentary assessment, resilience

## Abstract

The effects of stress may be alleviated when its impact or a decreased stress-resilience are detected early. This study explores whether wearable-measured sleep and resting HRV in police officers can be predicted by stress-related Ecological Momentary Assessment (EMA) measures in preceding days and predict stress-related EMA outcomes in subsequent days. Eight police officers used an Oura ring to collect daily Total Sleep Time (TST) and resting Heart Rate Variability (HRV) and an EMA app for measuring demands, stress, mental exhaustion, and vigor during 15–55 weeks. Vector Autoregression (VAR) models were created and complemented by Granger causation tests and Impulse Response Function visualizations. Demands negatively predicted TST and HRV in one participant. TST negatively predicted demands, stress, and mental exhaustion in two, three, and five participants, respectively, and positively predicted vigor in five participants. HRV negatively predicted demands in two participants, and stress and mental exhaustion in one participant. Changes in HRV lasted longer than those in TST. Bidirectional associations of TST and resting HRV with stress-related outcomes were observed at a weak-to-moderate strength, but not consistently across participants. TST and resting HRV are more consistent predictors of stress-resilience in upcoming days than indicators of stress-related measures in prior days.

## 1. Introduction

Stress is associated with an increased risk of numerous diseases [1,2,3,4,5,6,7] and mental disorders [8,9]. Besides these adverse effects on individuals, it also imposes a large financial burden on society via absenteeism, healthcare costs, and productivity loss [10,11]. Personalized just-in-time interventions may be able to prevent or alleviate some of these burdens [12]. To do this, either the negative impact of stress or a decreased resilience to cope with stress should be detected early, preferably via unobtrusive monitoring. For instance, unobtrusive detection of the negative impact of stress (e.g., on sleep or physiological systems) in an early state may help increase awareness that current circumstances may be causing wear and tear on bodily systems (allostatic load) that may be contributing to health-related problems if sustained over time [13]. Similarly, recognition of the potential depletion of resources that are needed for resiliently coping with challenges could be used to trigger feedback to take it easy that day and avoid overly challenging circumstances where possible. Recent developments in wearable sensor technology introduce promising opportunities for this type of unobtrusive monitoring [14,15].

When the first modern wearables came to market around 2009 (e.g., the Fitbit Classic), these devices initially became popular as pedometers or activity trackers but were already able to estimate sleep duration via accelerometry as well [16]. Since then, consumer wearable-based sleep tracking has improved to a point where it is considered proficient for measuring the Total Sleep Time (TST), while the detection of sleep stages needs further work [17]. Sleep deprivation is known to have a reciprocal relationship with stress, meaning that it is both caused and can be caused by stress [18]. Longitudinal studies with repeated daily measures confirm this bidirectional association [19,20,21] but tend to rely on subjective TST measures (e.g., measured via questionnaires) and need verification using objective sleep measurements [22]. Wearable-based research can therefore contribute to this body of knowledge and explore the potential of wearables to unobtrusively monitor for signs of the negative impact of stress or decreased resilience.

Besides behavioral outcomes such as physical activity and sleep, around 2015 (e.g., the Fitbit Charge HR) consumer wearables started measuring heart rate after photoplethysmography (PPG) sensors were included [23]. Today, PPG sensors are also used to track physiological outcomes such as heart rate, blood oxygen saturation, blood pressure, and respiration [24]. Perhaps the most important PPG-based innovation in the context of stress and resilience is the measurement of Heart Rate Variability (HRV), which can now be accurately measured using wearables or even camera-based smartphone apps in a resting state or during sleep [25]. HRV is a measure of the variation in heartbeats and is a proxy for autonomous nervous system functioning [26]. HRV acutely declines during stress [27] and afterward can remain suppressed during subsequent sleep [28,29]. Consequently, individuals with a low resting HRV are more likely to interpret seemingly mild stimuli as significant stressors [30,31,32] and have suboptimal emotion regulation [33,34]. Although these findings are based on population studies that investigated between-subject differences, the reciprocal nature of these findings illustrates that an initial decline in resting HRV could potentially cascade into subsequent days and thus have downstream effects.

A recent paper introduced a conceptual model in which the potential underlying mechanism for such a cascading effect of an initial decline in resting HRV was described [14]. The model suggests that resting HRV buffers against the impact of demands on stress by making potentially stressful situations seem less stressful [30,31,32], as well as against the impact of stress on mental exhaustion via more optimal emotion regulation [33,34]. Since this model also proposes that the need for recovery (e.g., increased mental exhaustion and/or decreased vigor) negatively influences resting HRV [28,29], a potential negative feedback loop is formed. This aligns with the conservation of resources theory, which states that since resources are needed to cope with demands, an initial loss of resources may result in a loss spiral [35]. Finally, the model hypothesizes stress to both be negatively impacted by stress [18,19,20,21], as well as to buffer against the negative impact of an increased need for recovery on resting HRV due to its restorative properties [36,37]. A study was then performed to test these hypotheses by utilizing wearables to measure TST and resting HRV, as well as an Ecological Momentary Assessment (EMA) smartphone app to measure subjective demands, stress, and mental exhaustion [38]. The study confirmed that resting HRV is both negatively impacted by mental exhaustion and buffers against the negative associations between demands and stress, as well as stress and exhaustion. Day-to-day changes in resting HRV may therefore be both indicative of the negative impact of stress and predictive of stress-resilience, potentially even on a multi-day level. Further exploration of these potential multi-day bidirectional associations will improve our understanding of the degree to which day-to-day changes in wearable-measured resting HRV can be interpreted as potentially stress-related and in which they should be expected to reflect a state of lowered resilience.

To summarize: wearable-measured sleep and resting HRV have both been bidirectionally associated with subjective stress-related outcomes, but within-subject research investigating the potential patterns in multi-day associations in a real-world context is lacking. Increased insight into the degree to which these relationships are consistently observed in individuals may help improve models for the early recognition of the negative impact of stress and of lowered resilience. Such insights could contribute to the development of automated resilience interventions that may help to prevent stress-related problems. These interventions are especially relevant for individuals working in safety-critical professions, such as police officers [39]. Therefore, this study explores whether wearable-measured TST and resting HRV in police officers (1) can be predicted by stress-related EMA outcomes (demands, stress, mental exhaustion and vigor) in the preceding days, and (2) predict stress-related EMA outcomes in the subsequent days.

## 2. Materials and Methods

### 2.1. Study Design

An observational multiple n-of-1 study design was used [40], where individuals collected data on a daily basis, which were then individually assessed as independent time series. The results are therefore presented as a series (n = number of participants) of independent quantitative analyses on within-subject associations (e.g., a case series) based on samples with a high number of observations per participant (N = number of observations per participant) that can be relatively well-intercompared due to consistency in the applied methods. The current design is therefore optimized to provide a first exploration of possible multi-day associations at a within-subject level based on high-quality data, as is the aim of this study. Additionally, a rough estimate of the extent to which the respective associations may be found across individuals can be described in order to guide future studies. The current methods were based on a prior study that investigated nested within-day associations [14,38]. Missing data are problematic for time-series analysis. To limit missing data, we made several adaptations to optimize the previously used research design. We included automatic resting HRV measurements, a shorter daily Ecological Momentary Assessment (EMA) questionnaire, and an improved reward for adhering to the measurement protocol (participants were allowed to keep the wearable if they collected at least 100 complete daily observations). Data were collected for two purposes: (1) comparing longitudinal (5-week) trends in daily resting HRV and fluctuations therein to full questionnaire outcomes for stress, somatization, anxiety, and depression, and (2) the assessment of potential bidirectional and/or multi-day associations of sleep and resting HRV with stress-related EMA outcomes. The results of the former are published elsewhere [41], whereas the results of the latter are presented in this paper. The study protocol was approved by the ethical committee of the Hanze University of Applied Sciences Groningen (heac.2020.012).

### 2.2. Participants

Police officers working in a large Dutch city and possessing an Android- or iOS-based smartphone were invited to participate by the human resources bureau of their office. Interested respondents received the study information via e-mail. Participation was voluntary. The data collection period lasted a minimum of 15 weeks but could be extended with a number of additional 5-week periods. Extending the data-collection period was optional. Since at least 20 but preferably 50 observations with limited missing data are needed for accurate time series analysis [42,43], this data collection period (105–140 days) was expected to be appropriate to collect sufficient data. To also minimize potential missing data participants could keep the wearable and received a personal feedback report after the study as a reward if they collected complete daily data for at least 100 days and completed all baseline and 5-weekly questionnaires. The recruitment period lasted until the maximum capacity of 10 participants was reached (i.e., it ran from June 2020 and July 2020). Before the start of their data collection, participants had a conversation with the first author during which the study requirements were explained, and participants gave their written informed consent. Due to COVID-19 restrictions, all contact with the participants occurred via teleconferencing and e-mail. After data collection, one participant was excluded because they were diagnosed with atrial fibrillation. This participant’s data were excluded from the study, because this may have interfered with the accuracy of the HRV measurements. Another participant of whom only 56.3% of the daily observations were available was also excluded from the analysis. The remaining eight analyzed participants were predominantly male (n = 6), had an average age of 37.0 years (range: 29–51), and contributed at least 80% (range: 80.7–96.8%) of complete observations.

### 2.3. Data-Collection

#### 2.3.1. Baseline Questionnaires

Immediately after consent was provided, participants were asked to fill in a baseline questionnaire. The baseline questionnaire included two items on gender and birthdate, as well as full questionnaires on personality traits (the Big Five Inventory; BFI) [44], symptoms of distress, somatization, depression, and anxiety (the Four-Dimensional Symptom Questionnaire; 4DSQ) [45], burnout (the Oldenburg Burnout Inventory; OLBI) [46] and work engagement (the Utrecht Work Engagement Scale; UWES) [47]. The outcomes of the baseline questionnaires and the mean values of the daily wearable and EMA outcomes were summarized per participant and on aggregate and used to describe the current sample for generalization purposes and as background information on the characteristics of the participants (Table 1). The age and gender of the individual participants were not described out of privacy considerations.

#### 2.3.2. Wearable-Based Variables

The Oura ring (generation 2, Oura Ring, Oulu, Finland) was used to measure TST and resting HRV during sleep. The Oura ring is a consumer wearable that measures sleep, physical activity, temperature, heart rate, and HRV. The consumer-available ring contains 2 infrared Light-Emitting Diode (LED) sensors, 2 Negative Temperature Coefficient (NTC) thermistor sensors, a tri-axial accelerometer, and a gyroscope. Although the algorithms that are used by the Oura ring to classify sleep and HRV based on the outputs of these sensors are proprietary, the ring (generation 2) has been confirmed to provide valid measurements of TST [48,49] and HRV [25,50,51] in independent research. Participants used a ring-size kit to determine their correct ring size to optimize fit for both user comfort and measurement accuracy and were allowed to choose a ring color of their preference. To preserve privacy, anonymized Oura accounts were created by using e-mail addresses on a custom domain to create accounts without the participants’ names. The Oura-reported TST was used, which represents the total Duration of the Sleep Episode (DSE) minus the Sleep Onset Latency (SOL) and Wake-time After Sleep Onset (WASO). Similarly, the Oura-reported HRV was used, which represents the root Mean Square of the Successive Differences (rMSSD) in the inter-beat-intervals. This metric was then logarithmically transformed (lnrMSSD) to improve its distribution for statistical modeling, which is a common procedure in HRV research [52]. Finally, the Moderate-to-Vigorous Physical Activity (MVPA) was used as a control variable during analysis [53].

#### 2.3.3. Ecological Momentary Assessment-Based Variables

Every day at 7 PM, participants received a notification that a new EMA questionnaire was available on their smartphone app. Participants were instructed to complete the EMA before they went to bed. Since participants regularly worked night shifts, the EMA was available until 3 PM on the next day while participants received a reminder at noon to fill in their previous-day questionnaire if they had not finished it already. The EMA items were based on items used in a similar study [14,38]. The EMA measured: demands (*“How demanding was your day?”*), stress (*“How much stress did you perceive today?”*), mental exhaustion (*“I felt mentally exhausted as a result of my activities”*), vigor (*“Do you feel like undertaking activities?”*), and alcohol intake (*“I consumed … alcoholic beverages today”*). The demands, stress, and vigor items were scored on an 11-point Numeric Rating Scale (NRS), ranging from 0 (*“Not at all”*) to 10 (*“Extremely”*). Mental exhaustion was scored on an 11-point NRS ranging from 0 (*“Strongly disagree”*) to 10 (*“Strongly agree”*). The item for stress was based on a validated single-item scale [54], the item for mental exhaustion on an item of the Need For Recovery Scale [55], the item for vigor on an item of the Utrecht Work Engagement Scale [47], whereas the item for demands was self-composed in a similar style as the item for stress. The number of alcoholic beverages participants consumed during the passing day was included for use as a control variable during analysis and based on the AUDIT-C questionnaire [56], since alcohol consumption is known to impact resting HRV [57].

### 2.4. Data-Analysis

All analyses were performed in RStudio version 2022.7.1.554 [58] using R version 4.2.1 [59]. The ‘zoo’ package was used for linear interpolation of missing data [60], the ‘vars’ package was used for Vector Auto-Regression (VAR) modeling, Granger causation testing, and Impulse Response Function (IRF) calculation [61]. Finally, ‘ggplot2′ was used to visualize the IRFs [62].

#### 2.4.1. Data Preparation

First, descriptive statistics on the number of observations, the percentage of complete observations, and the wearable- and EMA-variables were calculated based on the full set of collected data. Since VAR analyses do not allow for missing data, missing data were imputed via linear interpolation. Rows, where data were missing at the beginning or end of the time series, were removed, as these could not be imputed. All values were standardized (by first subtracting the within-subject mean from each daily value and then dividing it by the within-subject standard deviation) to optimize the inter-comparability of beta-coefficients and prevent multicollinearity. Finally, two versions of the vectors with the four core EMA variables (demands, stress, mental exhaustion, and vigor), two core wearable variables (TST and resting HRV), and two control variables (MVPA and alcohol consumption) were constructed to answer both research questions. The vector for the first analysis contained rows with values for the passing night’s TST and nocturnal HRV, combined with the EMA items of the subsequent evening so that the lagged values of the EMA items (the values on the previous row that represent the EMA of the previous day) could be interpreted as predictors for TST and HRV (the values on the current row that represent the values for the passing night). For analysis 2, the TST and HRV data were shifted to the previous day, so that the lagged values of the TST and HRV (the values on the previous row, representing the passing night) could be interpreted as predictors for the EMA items (the values on the current row, representing the current day).

#### 2.4.2. Vector Auto-Regressive Modeling

To assess the stationarity of the time series as a prerequisite to performing VAR analysis, the Phillips-Perron (PP) unit root test was used on all variables [63]. All-time series were stationary (PP *p* < 0.05). Next, the number of lags (i.e., number of preceding days included as predictor values) to include in the VAR model was determined. This was completed via the ‘VARselect’ function, which calculates models up to 7 lags (i.e., one full week’s worth of lags). The most optimal lag order is based on four information criteria corresponding to the different models (i.e., Akaike Information Criterion (AIC), Hannan-Quinn (HQ) criterion, Schwarz Criterion (SC), and Final Prediction Error (FPE) criterion). The mode of these four information criteria was selected as the most optimal lag order used in the VAR model. In the case of a tie, the most conservative estimate was chosen. Assumptions were tested on the residuals of the VAR model. The residuals were assessed for autocorrelation via an asymptotic multivariate Portmanteau Test (PT) [64], for heterogeneity via an ARCH-LM test [65], and for normality via a Jarque-Bera (JB) test [66].

#### 2.4.3. Granger Causation Testing

To increase confidence in the predictive value of core EMA variables that were found to be statistically significant predictors of wearable variables (or vice versa) in the full VAR models, Granger causation tests were applied [67]. Granger causation tests assess if the inclusion of a predictor significantly improves a VAR. To isolate the direct relationships between these associations of interest from interrelations with the other variables in the vector, the Granger causation tests were applied to vectors with only the core predictor and outcome variables. Therefore, significant Granger causation tests showed that the predictor variable itself explains meaningful variance in the outcome variable and is not just significant in the VAR due to interrelations with other variables in the vector.

#### 2.4.4. Impulse Response Function Visualization

An IRF is the reaction of a dynamic system in response to an external change [68]. IRF visualizations of relevant predictors on the outcomes can illustrate how the outcome varies on subsequent days after being faced with an increase in a predictor variable. The IRF of predictors that were both statistically significant in the full VAR model and in the additional Granger causation test were visualized. The IRF visualizations consisted of an overlay of participants where the respective association was observed. The IRFs with the same predictor were grouped in a grid in order to cluster visualizations of the multi-day impact of a predictor on all relevant outcomes (including bootstrapped 95% confidence interval (CI) based on 1000 runs).

## 3. Results

### 3.1. Participant Characteristics

The eight participants, who were 29.4 to 51.1 years old (median = 36.8) and predominantly male (75%), collected 125 to 386 observations per person (median = 147) of which 80.7 to 96.8% (median = 90.7%) contained complete data on the EMA outcomes, as well as daytime and nighttime wearable outcomes. The average TST ranged from 5.5 to 7.6 h (median = 6.8), during which they had an average resting HRV (rMSSD) of 26.8 to 72.8 milliseconds (median = 45.6). The participants were moderate-to-vigorously physically active for 18.5 to 54.4 min per day (median = 39.4). The median reported daily scores on the stress-related outcomes was in the lower half of the scale (0–10) for demands (median = 4.2, range = 2.4–5.1), stress (median = 2.7, range = 1.3–3.9) and mental exhaustion (median = 3.0, *range* = 1.4–4.8). The mean reported daily scores on vigor were in the upper half of the scale (median = 5.6, range = 4.9–7.4). On average, the participants consumed between 0.1 and 0.9 (median = 0.4) alcoholic beverages per day. An overview of all participant characteristics is presented in Table 1.

### 3.2. Analysis 1: Predicting TST and HRV by EMA

All analyzed time series were found to be stationary (PP unit root test *p* < 0.05). The AIC, HQ, SC, and FPE information criteria that were used to determine the lag order for the VAR models unanimously suggested an optimal lag order of 1 in all participants, with exception of participant 7, where 2 out of 4 information criteria suggested a lag order of 2. Since the conservative option was chosen in case of a tie (§2.4.2), VAR models with 1 lag were created for all participants. No heterogeneity (ARCH-LM test *p* > 0.05) was found in the residuals of any model. The residuals also contained no autocorrelation (PT *p* > 0.05) in most participants, except for participant 5. This autocorrelation could not be resolved (e.g., by adding additional lags to the model), and suggests that an unobserved but relevant factor was not included in the model, which therefore may be useful but not complete. Finally, none of the residuals of any model were found to be normally distributed (JB-test *p* < 0.05). This was likely attributable to the distribution of some of the EMA items, which were occasionally skewed or even bimodal. Since simulation studies showed that non-normally distributed residuals are not problematic in analyses with a sample of at least 100 observations [69], this was not considered to be a problem for the interpretation of these results.

The results of the VAR models on TST are presented in Table 2. Demands was a statistically significant (*p* < 0.05) negative predictor of TST for three participants (4, 5, 7). For participant 5 this finding was confirmed by a statistically significant Granger causation test. Mental exhaustion was a significant positive predictor of TST in participant 4, but this was not confirmed in Granger causation testing and therefore interpreted as a potentially spurious relationship. Stress and vigor were not statistically significant predictors of TST in any model. The explained variance in the TST of the participant (5) in which demands was a significant predictor that was confirmed by a significant Granger causation test was 9%.

The results of the VAR models on HRV are presented in Table 3. Demands was a significant negative predictor of HRV in participant 3, which was also confirmed via Granger causation testing. Mental exhaustion was a significant positive predictor of HRV in participant 4, but it was in Granger causation testing and therefore interpreted as a potentially spurious relationship. Stress and vigor were not statistically significant predictors of HRV in any model. The explained variance in the HRV of the participant (3) in which demands was a significant predictor that was confirmed by a significant Granger causation test was 22%.

To support the interpretation of the temporal associations where both the beta-coefficient and Granger causation tests were significant, IRF visualizations for the impact of an increase in demands on (A) TST and (B) HRV are displayed in Figure 1. In both outcomes, an increase in demands results in a sudden drop in the outcome variable, which then gradually recovers in subsequent days. However, the recovery of HRV takes longer (0 enters the 95% CI on the sixth day) than that of TST (0 enters the 95% CI on the third day). This difference can be attributed to the highly significant autoregression component in HRV (*p* < 0.001), which is not observed in TST. This means that resting HRV values are relatively likely to be similar to the previous day (e.g., if yesterday’s resting HRV value was relatively low, today’s value is likely to be relatively low again), whereas TST values have little to no association to the value of the previous day. The impact of demands, therefore, appears to be more long-lasting on HRV than on TST—at least in these participants.

### 3.3. Analysis 2 Predicting EMA by TST and HRV

The outcomes of the pre- and post-model diagnostic tests of analysis 2 were similar to those of analysis 1. The only difference in the pre- and post-model diagnostic tests of analysis 1 is that in analysis 2, participant 7 had just 1 out of 4 information criteria suggesting an optimal lag order of 2 instead of 2 out of 4. Therefore, VAR models with 1 lag were again created for all participants.

The results of the VAR models on demands are presented in Table 4. TST was a statistically significant negative predictor of demands in two participants (1, 2), which was confirmed with the Granger causation test in both cases. HRV was a significant negative predictor of demands in two participants (7, 8), also confirmed via Granger causation tests. The explained variance in the demands of these participants was 16% and 23%, respectively.

Table 5 contains the results of the VAR models on stress. TST was a significant negative predictor of stress in three participants (1, 2, 7), all confirmed via the Granger causation test. HRV was a significant negative predictor of stress in one participant (8), again confirmed via Granger causation tests. The explained variance in the stress of these participants ranged from 2% to 23%.

The results of the VAR models on mental exhaustion are presented in Table 6. TST was a significant negative predictor of mental exhaustion in five participants (1, 2, 3, 5, 7), all confirmed via Granger causation tests. HRV was a significant negative predictor of mental exhaustion in one participant (8), again confirmed via Granger causation testing. The explained variance in the mental exhaustion of these participants ranged from 3% to 36%.

Finally, the results of the VAR models on vigor are presented in Table 7. TST was a significant positive predictor of vigor in five participants (1, 3, 4, 5, 7), all confirmed via Granger causation tests. HRV did not predict vigor in any participant. The explained variance in the vigor of these participants ranged from 8% to 34%.

IRF visualizations for the impact of an increase in TST on each of the four EMA outcomes (A–D) are displayed in Figure 2. In all outcomes, an increase in TST resulted in a sudden decline (or incline in the case of vigor) that recovered (0 enters the 95% CI) in the subsequent 1 or 2 days. The IRF visualizations for the impact of an increase in HRV on the four EMA outcomes (Figure 3A–C) is similar for demands (1–2 days), although recovery from the impact on stress (2–3 days) and mental exhaustion (2–3 days) appears to take a bit longer. It appears that in these participants, the impact of changes in HRV is more long-lasting than for changes in TST.

## 4. Discussion

This study aimed to explore to what degree wearable-measured sleep and resting HRV in police officers (1) can be predicted by stress-related EMA outcomes in the preceding days, and (2) predict stress-related EMA outcomes in the subsequent days. After performing a time series analysis on eight participants, the results showed that associations in both directions of modest strength were observed and that TST and resting HRV were more consistent predictors for the next day’s perceived demands, stress, mental exhaustion, and vigor than the other way around. Demands was a negative predictor of TST of one participant, and for resting HRV in another. Mental exhaustion predicted both resting HRV and TST in the same participant. Especially, TST seemed a strong predictor of stress-related EMA outcomes. TST negatively predicted demands in two participants, stress in three participants, mental exhaustion in five participants, and positively predicted vigor in five participants. Resting HRV negatively predicted demands in two participants, and both stress and mental exhaustion in one participant.

This study led to three key findings that will first be reflected upon, followed by a discussion of the strengths and limitations of the study, and finally a summary of the main conclusions and recommendations for future research.

### 4.1. Associations between TST, HRV and EMA Outcomes Are Not Consistently Observed

Although TST was a negative predictor of mental exhaustion and a positive predictor of vigor in the majority (62.5%) of the participants, no association between a wearable- and an EMA-based item was consistently observed in all participants. No convincing explanations for the prevalence of these associations were identified after inspection of differences in the participant characteristics (Table 1).

The number of participants in this study (n = 8) was too low to meaningfully assess to what extent between-subject differences in participant characteristics could predict the prevalence of these associations. Future studies with a larger sample size are recommended to explore if the occurrence or strength of these associations may be explained by participant characteristics, for instance via multilevel VAR [70]. If these differences can be explained in future studies, they may be used to further personalize wearable-based models for stress-resilience.

It is also possible that the strength of these associations does not (only) depend on differences between individuals, but (also) on differences within individuals or in their environment. However, it may be difficult to determine beforehand what these influencing factors may be. It is possible to first explore if the strength of these relationships changes over time, for example via time-varying VAR models [71]. Detecting such changes over time is particularly feasible in datasets with a larger number of observations and/or more granular data. If these associations do change over time, it is possible that they may be actually relevant for all participants, but only under certain circumstances. Depending on the outcomes of such studies, it could provide new insights into the internal or external factors that determine when these associations are observed.

### 4.2. The Impact of Changes in HRV Appears to Be More Abiding than That of Changes in TST

The IRF visualizations in Figure 1 demonstrated that a demand-induced decline of resting HRV appears to have a longer recovery time (5–6 days) than a demand-induced decline of TST (2–3 days). Similarly, the impact of a change in resting HRV on stress-related EMA outcomes (Figure 3) appears to also be more long-lasting (1–3 days) than that of a change in TST (1 day) (Figure 2). This was attributed to the significant autoregression component that was observed in resting HRV, but not in TST. The strong autoregression component in the resting HRV model means that resting HRV values are relatively likely to be similar to those of the previous day(s). Therefore, a demand-induced decline in resting HRV (analysis 1) may take several days to recover from. Similarly, the impact of a decline in resting HRV on demands, stress, mental exhaustion, and vigor is likely to spill over into subsequent days, as it means that resting HRV is likely to remain suppressed for another few days.

This observation may be explained by the fundamentally different nature of the concepts resting HRV and TST. Resting HRV is a quantification of a physiological state that is continuously striving to maintain stability (homeostasis) despite disruptive challenges (allostasis) [13]. The recovery from a stressor that has a physiological impact (allostatic load) could take longer depending on the intensity and frequency of the stressor, as well as the quality and quantity of the subsequent recovery [36,37]. As such, a large decline in resting HRV can logically be expected to take some time as well. TST, on the other hand, is a quantification of the recovery process itself. Stress can negatively influence TST on the following night [18,19,20,21] and can therefore also impact TST on subsequent nights in the case of a recurring or sustained stressor. However, when this is not the case, it is also possible that the individual compensates for the previous sleep loss via recovery sleep [72], which would mean that TST on a subsequent night is no longer suppressed but actually increased. From this perspective, TST values can be expected to be more volatile than changes in resting HRV and thus have a weaker autoregression component. However, it is possible that changes in TST do have a longer-lasting impact on relevant underlying (psycho)physiological states such as vigor, which was observed to consistently have a significant autoregression component (Table 4).

The seemingly more abiding impact of a change in resting HRV on the resting HRV of the subsequent days may also be influenced by the development of a negative feedback loop. A previous study showed that evening mental exhaustion negatively impacted subsequent resting HRV and that resting HRV itself buffered against the positive association between demands and stress, as well as between stress and mental exhaustion [38]. This aligns with the Conservation of Resources Theory, which describes that an initial loss of resources could lead to a negative feedback loop. This means that fewer resources are available to handle upcoming challenges, which leads to lower resilience [35]. However, in the current study, no bidirectional association between a stress-related EMA item and resting HRV was observed within a single participant. Future studies with a larger sample are needed to increase insight into the multi-day impact of stress-related changes in resting HRV.

### 4.3. TST and HRV Are More Consistent Predictors of Stress-Related Outcomes than Vice Versa

These findings indicate that wearable-measured TST and HRV seemed better predictors of stress-related EMA outcomes than the other way around. EMA-based predictions of TST and resting HRV were only observed in two participants, who had relatively large samples of observations (N = 385 and N = 283) compared to the median (N = 144). Additionally, these relationships were not consistently observed in both participants. These differences cannot merely be explained by statistical power. Nevertheless, these models explained a modest amount of variance in TST (9%) and resting HRV (22%) in some participants. It is possible that these relationships are relatively small in nature and can only be observed in larger samples.

The finding that TST is a more consistent predictor of stress-related outcomes than that it can be predicted by stress-related outcomes aligns with prior research [22]. For instance, a lower TST has consistently been shown to predict increased stress [19,20,21,73]. Conversely, in the same studies, the opposite is regularly associated with smaller effect sizes [19,21], but in another study, TST was not associated with stress-related outcomes [73]. 

Similar scientific findings on the combination of both the predictive power and predictability of resting HRV in the context of stress-related outcomes are limited. However, the current findings do align with prior research, which has shown that stress-related outcomes negatively affect resting HRV [27,28,38] and that a relatively lower resting HRV than an individual’s normal resting HRV can negatively impact stress-related outcomes on the following day [38,74].

One of the implications of this finding is that a decrease in wearable-measured TST or resting HRV does not necessarily point toward the occurrence of stress-related outcomes. Although the observed decrease in TST or resting HRV might have been caused by subsequent high demands or stress, this outcome may have been confounded by other factors. In situations where sudden extreme demands or stress occur, this might in some cases directly cause a decreased TST or resting HRV. However, in these circumstances, the wearable-user is likely already aware of the impact of such events. In such instances, the wearable-user less likely needs objective feedback to confirm this short-term effect.

Based on these findings, wearable-measured TST and resting HRV are not necessarily usable as a direct indication of the negative impact of stress but hold more promise to function as potential predictors to estimate one’s resilience. For instance, these insights could be implemented in resilience interventions in the form of a readiness score that gives the user feedback on his or her expected readiness to handle mental and physical challenges that day [75]. Ideally, these factors will be expanded upon in future research (e.g., by also assessing behavioral outcomes such as smartphone usage, geolocation, or patterns in communication) that also explore different modeling approaches (e.g., machine learning) in order to improve the performance of these models.

### 4.4. Strengths and Limitations of the Current Study

This study applied a novel research design and recruited a motivated number of participants that resulted in a series (n = 8) of sizable datasets (N = 125–386) with mostly (80.7–96.8%) complete observations. By utilizing a consumer-available wearable that is validated for both TST and resting HRV measurements to collect observational data in a real-life environment, the generalizability of the findings to practical settings is relatively good. However, three limitations of the current study should also be considered during the interpretation of the presented results.

First, the multiple n-of-1 study design with a small number of participants (n) but a large number of observations per participant (N) was optimized as a first exploration of the potential existence of the hypothesized multi-day associations at a within-subject level based on high-quality data but limits the generalizability of the current findings to a broader target population. Therefore, future research with a larger number of participants is needed to increase confidence that the found associations are indeed relevant for larger groups of people. Future research is also needed to better understand why the identified relationships are prevalent in some cases, but not in others. For instance, it is possible that studies with a larger number of observations per participant can unveil to what extent associations with a smaller strength can be observed in other participants, and to what extent the strength of these associations may change over time (e.g., via time-varying VAR).

Second, the included healthy participants and data collection during the COVID-19 lockdown might have affected the participants’ perceptions of demanding and stressful situations. Their daily practice may not have been very demanding, which may have resulted in relatively low variance in the data. This aligns with the findings of a study on 2567 European police officers, which reported decreased strain during the pandemic [76]. The analyzed participants all scored relatively well on the mental well-being questionnaires (Table 1). Another article that was based on data from this same study population showed that some participants reported moderately elevated stress and somatization throughout the study period, but that there were no clinically relevant signs of anxiety and depression [41]. Future studies with a more mentally challenged sample need to verify the current findings for more challenging conditions.

Finally, some of the statistical assumptions of the created VAR models were technically violated. Most notably, none of the VAR models had normally distributed residuals, which was likely the result of sometimes skewed or bimodally distributed EMA items. Since simulation studies have shown that this assumption is particularly relevant when relatively small samples are assessed but not problematic when a sample of at least 100 observations is analyzed [70], this was not considered to be a problem for the interpretation of the results. The VAR model of participant 5 was also found to have autocorrelated residuals, which could not be resolved (e.g., by adding additional lags). Although this does not necessarily limit the interpretability of the model and related findings, it does show that the model is incomplete, and at least one unobserved but relevant factor was not included in the present study.

## 5. Conclusions

This multiple n-of-1 study showed that in relatively healthy police officers, demands were occasionally observed to be a negative predictor of wearable-measured TST and resting HRV. TST and resting HRV were more regularly observed to be negative predictors of demands, stress, or mental exhaustion, whereas TST also positively predicted vigor in several participants. The presented results illustrate that caution is needed when interpreting changes in TST and resting HRV to be potentially stress-related and that TST and resting HRV are more likely to be useful as predictors of stress-resilience (e.g., expressed as a readiness score).

However, since the identified associations were not consistently observed amongst participants, further research is necessary to better understand the underlying mechanism. For instance, future studies with a larger sample of participants, which is also needed to improve the generalizability of the current findings, could consider assessing if these between-subject differences could be explained by participant characteristics (e.g., via multilevel VAR). Another direction could be to explore if the strength of these associations’ changes over time in samples with a larger number or more granular data (e.g., via time-varying VAR). Finally, future studies should explore if predictive models with a higher explained variance can be achieved by including additional data sources (e.g., smartphone usage, geolocation, or patterns in communication) or utilizing more inductive methods (e.g., machine learning approaches).

## Figures and Tables

**Figure 1 sensors-23-00332-f001:**
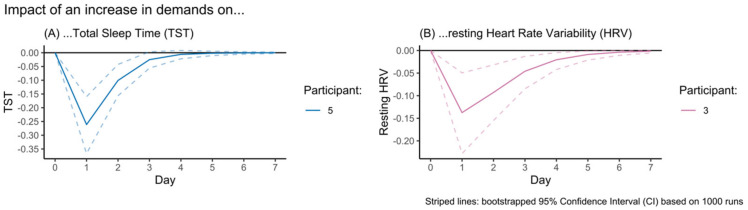
Visualization of the Impulse Response Function (IRF) of the impact of an increase in demands on the Total Sleep Time (TST) and resting Heart Rate Variability (HRV) during the subsequent week.

**Figure 2 sensors-23-00332-f002:**
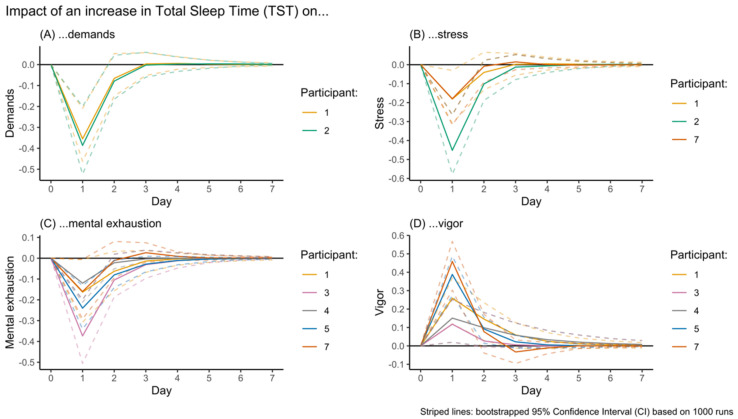
Visualization of the Impulse Response Function (IRF) of the impact of an increase in Total Sleep Time (TST) on the subsequent week’s demands, stress, mental exhaustion, and vigor.

**Figure 3 sensors-23-00332-f003:**
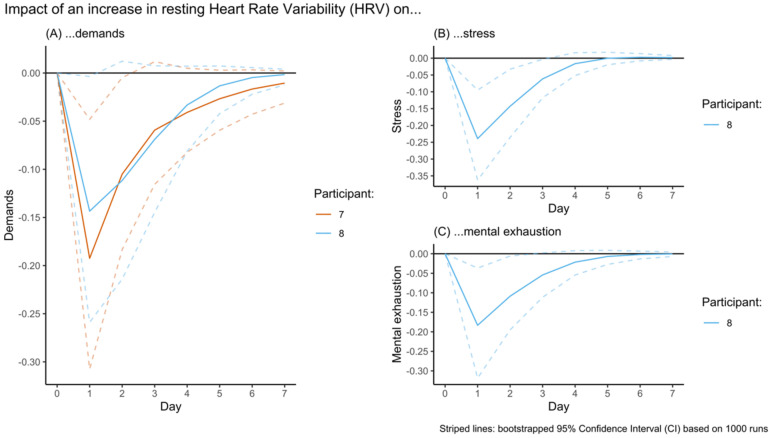
Visualization of the Impulse Response Function (IRF) of the impact of an increase in resting Heart Rate Variability (HRV) on the subsequent week’s demands, stress, and mental exhaustion.

**Table 1 sensors-23-00332-t001:** Participant characteristics.

	Participant							
Baseline Questionnaires	1	2	3	4	5	6	7	8
Extraversion (1–5)	2.8	3.3	3.3	2.5	3.8	4.4	3.8	4.1
Agreeableness (1–5)	3.3	3.9	3.8	3.6	3.4	3.8	3.9	3.8
Conscientiousness (1–5)	3.9	3.8	3.3	3.3	3.8	3.0	3.8	4.0
Neuroticism (1–5)	2.1	2.5	2.4	2.4	2.5	2.3	2.9	2.4
Openness (1–5)	4.1	2.8	3.7	3.6	3.4	3.8	2.5	3.4
Distress	Low	Moderate	Low	Low	Low	Low	Moderate	Low
Depression	Low	Low	Low	Low	Low	Low	Low	Low
Anxiety	Low	Low	Low	Low	Low	Low	Low	Low
Somatization	Low	Low	Low	Low	Low	Low	Low	Low
Exhaustion (1–4)	3.0	2.5	3.3	2.5	3.0	2.6	2.6	2.6
Disengagement (1–4)	2.4	2.6	3.3	2.9	2.9	2.3	2.4	3.8
Work engagement (1–7)	5.4	4.6	5.3	4.5	5.2	5.1	4.2	5.9
**Daily Measurements**								
Total observations	147	125	386	150	285	143	147	140
% complete observations	81.6	94.4	88.3	80.7	96.8	93.0	95.9	80.7
TST (hours)	7.0 (1.5)	6.5 (0.8)	7.4 (1.5)	5.5 (1.3)	6.7 (1.3)	7.6 (1.5)	7.4 (1.3)	6.5 (1.2)
HRV (milliseconds)	51.3 (15.6)	43.8 (5.1)	54.8 (15.9)	72.8 (9.8)	47.4 (15.2)	29.6 (4.3)	39.6 (6.9)	26.8 (5.1)
Demands (0–10)	4.4 (3.0)	4.7 (2.0)	3.2 (1.6)	5.1 (2.7)	4.1 (1.2)	2.4 (1.9)	4.3 (2.7)	3.4 (2.2)
Stress (0–10)	3.9 (2.7)	3.5 (1.9)	2.6 (1.8)	3.3 (2.0)	1.5 (0.9)	1.6 (1.8)	1.3 (1.9)	2.8 (1.6)
Mental exhaustion (0–10)	2.8 (2.7)	4.8 (2.0)	3.7 (1.7)	4.2 (2.4)	2.1 (1.0)	1.4 (1.9)	2.6 (2.9)	3.3 (2.1)
Vigor (0–10)	5.3 (2.8)	5.6 (1.4)	5.8 (2.1)	5.0 (2.1)	4.9 (1.1)	6.0 (1.7)	5.5 (2.5)	7.4 (0.8)
MVPA (minutes)	54.2 (33.4)	49.3 (24.2)	45.4 (35.6)	28.0 (26.5)	49.7 (28.7)	33.4 (30.3)	25.7 (19.2)	18.5 (14.2)
Alcohol use (units)	0.4 (0.8)	0.5 (0.9)	0.4 (0.7)	0.1 (0.6)	0.5 (0.9)	0.1 (0.4)	0.9 (1.3)	0.1 (0.5)

Note. For the baseline questionnaires, the observed values are reported. For the daily measurements, mean and standard deviation are reported. TST: Total Sleep Time; HRV: Heart Rate Variability; rMSSD: root Mean Square of the Successive Differences; MVPA: Moderate-to-Vigorous Physical Activity.

**Table 2 sensors-23-00332-t002:** Vector autoregression models for Total Sleep Time (TST) per participant (1–8).

	TST							
	1	2	3	4	5	6	7	8
Independent Variable	β	β	β	β	β	β	β	β
Constant	0.00	0.03	−0.01	−0.00	0.01	−0.02	0.02	0.00
TST (lag 1)	0.08	0.06	−0.02	0.21 **	0.04	−0.10	−0.08	−0.05
HRV (lag 1)	0.07	0.01	−0.07	0.10	−0.10	0.04	−0.12	−0.08
MVPA (lag 1)	0.02	−0.09	0.05	0.18 *	0.04	−0.02	0.07	0.07
Alcohol intake (lag 1)	0.17	0.08	0.09	0.02	0.08	0.04	0.08	0.01
Demands (lag 1)	0.24	−0.02	0.05	−0.40 **	−0.39 ***	0.01	−0.21 *	0.02
Stress (lag 1)	−0.16	−0.13	−0.12	0.03	0.08	−0.07	−0.00	−0.12
Mental exhaustion (lag 1)	0.24	0.14	0.02	0.32 *	0.01	−0.14	0.10	−0.07
Vigor (lag 1)	−0.01	−0.06	−0.10	−0.10	0.10	0.16	−0.07	0.02
*N*	*142*	*123*	*385*	*148*	*283*	*141*	*146*	*138*
*Adjusted R^2^*	*0.03*	*−0.04*	*0.02*	*0.11*	*0.09*	*0.03*	*0.01*	*−0.03*
*F-statistic*	*1.49*	*0.48*	*1.76*	*3.26 ***	*4.53 ****	*1.47*	*1.19*	*0.54*

Note. *** *p* < 0.001, ** *p* < 0.01, * *p* < 0.05. *p* < 0.1; Underlined: both the beta-coefficient and Granger causation test *p* < 0.05. TST: Total Sleep Time; HRV: Heart Rate Variability; MVPA: Moderate-to-Vigorous Physical Activity.

**Table 3 sensors-23-00332-t003:** Vector autoregression models for Heart Rate Variability (HRV) per participant (1–8).

	HRV							
	1	2	3	4	5	6	7	8
Independent Variable	β	β	β	β	β	β	β	β
Constant	−0.01	0.02	−0.01	0.03	−0.01	−0.01	0.00	0.03
TST (lag 1)	0.04	0.00	−0.02	−0.03	−0.04	0.01	0.11	0.10
HRV (lag 1)	0.74 ***	0.06	0.40 ***	0.25 **	0.59 ***	0.30 ***	0.32 ***	0.36 ***
MVPA (lag 1)	0.03	0.03	−0.18 ***	0.10	−0.11 *	0.04	−0.14	−0.12
Alcohol intake (lag 1)	−0.12 *	0.06	−0.19 ***	−0.11	−0.18 ***	−0.10	−0.07	0.03
Demands (lag 1)	−0.01	0.01	−0.16 *	−0.20	0.01	0.16	−0.01	−0.08
Stress (lag 1)	0.05	0.03	−0.01	−0.22	−0.04	−0.23	0.04	−0.05
Mental exhaustion (lag 1)	0.05	−0.08	0.04	0.33 *	0.07	0.09	−0.07	−0.07
Vigor (lag 1)	0.05	−0.12	−0.02	0.15	−0.04	0.02	−0.06	0.06
*N*	*142*	*123*	*385*	*148*	*283*	*141*	*146*	*138*
*Adjusted R^2^*	*0.59*	*−0.05*	*0.22*	*0.12*	*0.40*	*0.07*	*0.12*	*0.17*
*F-statistic*	*26.61 ****	*0.34*	*14.20 ****	*3.46 ***	*24.15 ****	*2.41 **	*3.40 ***	*4.51 ****

Note. *** *p* < 0.001, ** *p* < 0.01, * *p* < 0.05. *p* < 0.1; Underlined: both the beta-coefficient and Granger causation test *p* < 0.05. TST: Total Sleep Time; HRV: Heart Rate Variability; MVPA: Moderate-to-Vigorous Physical Activity.

**Table 4 sensors-23-00332-t004:** Vector autoregression models for demands per participant (1–8).

	Demands							
	1	2	3	4	5	6	7	8
Independent Variable	β	β	β	β	β	β	β	β
Constant	−0.00	0.01	−0.01	0.03	0.00	−0.01	0.02	0.02
Demands (lag 1)	0.03	0.17	0.21 **	0.49 ***	0.34 ***	−0.09	0.21 *	0.31 **
Stress (lag 1)	0.25 *	−0.05	−0.06	−0.02	−0.17 *	0.17	−0.10	−0.06
Mental exhaustion (lag 1)	0.03	−0.03	0.08	0.06	0.08	0.18	0.05	0.13
Vigor (lag 1)	0.06	0.04	0.02	−0.06	0.15 *	0.03	−0.05	−0.04
MVPA (lag 1)	0.09	−0.06	0.08	0.00	0.12 *	0.06	0.01	0.19 *
Alcohol intake (lag 1)	−0.13	−0.07	−0.04	−0.10	−0.04	0.00	−0.24 **	−0.12
TST (lag 1)	−0.37 ***	−0.42 ***	−0.09	−0.06	0.00	−0.12	−0.11	0.08
HRV (lag 1)	0.06	0.14	−0.02	0.03	−0.07	0.11	−0.21 **	−0.18 *
*N*	*142*	*123*	*385*	*148*	*283*	*141*	*146*	*138*
*Adjusted R^2^*	*0.24*	*0.15*	*0.04*	*0.28*	*0.14*	*0.05*	*0.16*	*0.23*
*F-statistic*	*6.50 ****	*3.63 ***	*3.23 ***	*8.18 ****	*6.72 ****	*1.94*	*4.41 ****	*6.01 ****

Note. *** *p* < 0.001, ** *p* < 0.01, * *p* < 0.05. *p* < 0.1; Underlined: both the beta-coefficient and Granger causation test *p* < 0.05. MVPA: Moderate-to-Vigorous Physical Activity; HRV: Heart Rate Variability; TST: Total Sleep Time.

**Table 5 sensors-23-00332-t005:** Vector autoregression models for stress per participant (1–8).

	Stress							
	1	2	3	4	5	6	7	8
Independent Variable	β	β	β	β	β	β	β	β
Constant	−0.00	−0.00	−0.01	0.03	−0.00	0.01	−0.01	0.03
Demands (lag 1)	−0.03	0.01	−0.06	0.04	−0.02	−0.04	0.01	−0.10
Stress (lag 1)	0.32 *	0.06	0.22 **	0.49 ***	0.19 *	0.13	0.18	0.30 **
Mental exhaustion (lag 1)	0.02	0.06	0.05	0.01	0.07	0.14	−0.01	−0.10
Vigor (lag 1)	0.11	−0.03	−0.05	−0.00	0.11	0.06	0.01	0.00
MVPA (lag 1)	0.04	0.03	0.07	−0.07	0.09	0.01	−0.02	0.15
Alcohol intake (lag 1)	−0.18 *	−0.13	−0.08	−0.09	−0.02	−0.00	−0.01	−0.04
TST (lag 1)	−0.19 *	−0.49 ***	−0.04	0.01	−0.01	−0.05	−0.19 *	0.05
HRV (lag 1)	−0.12	0.08	0.03	−0.00	−0.02	0.08	0.06	−0.29 **
*N*	*142*	*123*	*385*	*148*	*283*	*141*	*146*	*138*
*Adjusted R^2^*	*0.18*	*0.23*	*0.04*	*0.23*	*0.05*	*0.01*	*0.02*	*0.12*
*F-statistic*	*4.88 ****	*5.44 ****	*3.17 ***	*6.56 ****	*2.80 ***	*1.09*	*1.39*	*3.32 ***

Note. *** *p* < 0.001, ** *p* < 0.01, * *p* < 0.05. *p* < 0.1; Underlined: both the beta-coefficient and Granger causation test *p* < 0.05. MVPA: Moderate-to-Vigorous Physical Activity; HRV: Heart Rate Variability; TST: Total Sleep Time.

**Table 6 sensors-23-00332-t006:** Vector autoregression models for mental exhaustion per participant (1–8).

	Mental Exhaustion							
	1	2	3	4	5	6	7	8
Independent Variable	β	β	β	β	β	β	β	β
Constant	−0.00	0.01	−0.01	0.02	−0.00	0.00	0.01	0.03
Demands (lag 1)	0.08	0.05	0.13	0.25 *	−0.03	−0.14	0.05	0.17
Stress (lag 1)	−0.07	−0.18	−0.17 *	0.17	0.02	0.22	0.08	0.02
Mental exhaustion (lag 1)	0.23	0.27 *	0.13	0.17	0.30 ***	0.04	0.16	0.01
Vigor (lag 1)	0.03	−0.07	−0.04	−0.14	0.00	−0.11	0.05	−0.05
MVPA (lag 1)	0.04	−0.06	0.01	−0.07	−0.05	0.08	−0.07	0.11
Alcohol intake (lag 1)	−0.17	−0.10	−0.02	−0.05	0.11	−0.05	−0.14	−0.09
TST (lag 1)	−0.17 *	−0.41 ***	−0.12 *	−0.05	−0.25 ***	−0.11	−0.17 *	0.02
HRV (lag 1)	0.10	0.16	−0.06	0.02	−0.00	0.04	−0.12	−0.22 *
*N*	*142*	*123*	*385*	*148*	*283*	*141*	*146*	*138*
*Adjusted R^2^*	*0.07*	*0.17*	*0.03*	*0.36*	*0.15*	*0.02*	*0.08*	*0.08*
*F-statistic*	*2.37 **	*4.05 ****	*2.68 ***	*11.15 ****	*7.04 ****	*1.44*	*2.61 **	*2.47 **

Note. *** *p* < 0.001, ** *p* < 0.01, * *p* < 0.05. *p* < 0.1; Underlined: both the beta-coefficient and Granger causation test *p* < 0.05. MVPA: Moderate-to-Vigorous Physical Activity; HRV: Heart Rate Variability; TST: Total Sleep Time.

**Table 7 sensors-23-00332-t007:** Vector autoregression models for vigor per participant (1–8).

	Vigor							
	1	2	3	4	5	6	7	8
Independent Variable	β	β	β	β	β	β	β	β
Constant	0.06	−0.00	−0.01	0.00	−0.00	−0.00	−0.01	0.02
Demands (lag 1)	−0.24 *	−0.10	−0.04	−0.16	−0.05	0.02	−0.09	0.07
Stress (lag 1)	0.32 **	−0.12	0.14 *	−0.03	0.09	0.04	−0.07	0.11
Mental exhaustion (lag 1)	−0.06	−0.16	0.04	0.01	−0.03	−0.14	0.02	−0.13
Vigor (lag 1)	0.37 ***	0.19 *	0.31 ***	0.48 ***	0.18 **	0.19 *	0.22 **	0.22 *
MVPA (lag 1)	−0.02	−0.04	0.02	0.05	0.06	0.05	0.10	0.13
Alcohol intake (lag 1)	0.11	−0.06	0.02	0.10	−0.04	0.12	0.10	−0.04
TST (lag 1)	0.27 ***	0.11	0.12 *	0.17 *	0.41 ***	0.13	0.49 ***	−0.17
HRV (lag 1)	0.06	−0.01	−0.01	−0.07	0.10	0.11	0.04	0.06
*N*	*142*	*123*	*385*	*148*	*283*	*141*	*146*	*138*
*Adjusted R^2^*	*0.27*	*0.11*	*0.08*	*0.34*	*0.20*	*0.08*	*0.34*	*0.05*
*F-statistic*	*7.45 ****	*2.93 ***	*5.23 ****	*10.60 ****	*9.62 ****	*2.44*	*10.39 ****	*1.81*

Note. *** *p* < 0.001, ** *p* < 0.01, * *p* < 0.05. *p* < 0.1; Underlined: both the beta-coefficient and Granger causation test *p* < 0.05. MVPA: Moderate-to-Vigorous Physical Activity; HRV: Heart Rate Variability; TST: Total Sleep Time.

## Data Availability

Data of this article are not publicly available due to the personal nature of the data in combination with the sensitive professions of the participants in this study.

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
