# Peer review of "Wearable-Measured Sleep and Resting Heart Rate Variability as an Outcome of and Predictor for Subjective Stress Measures: A Multiple N-of-1 Observational Study"

_sensors, 2022, doi:10.3390/s23010332_

Round 1
Reviewer 1 Report
This article reported the wearable-measured sleep and resting heart rate variability as a predictor for subjective stress measure. The author collected from the wearable device (Oura ring) altogether a series of questionnaires daily from voluntary participants who are police officers. The author used statistical tools to compare and find correlations among measured parameters to finally predict the stress-related outcome in subsequent days. As the author discussed the number of participants is a bit low, I do agree with this point and I think it would be a major issue, resulting in insufficient data to make a conclusion. Since total sleeping time and heart rate variability can be influenced by many other factors as the author mentioned, more data sets are required to be analyzed. Moreover, another concern is the questionnaires itself which are risky to be of bias and random. As the scales indicated in the questionnaires can be varied from one to another, to minimise the randomness, it requires an enormous set of data to observe the trend or reduce those issues. I agree that those autoregressive models can be used within limited data; however, we cannot deny the fact that the set of data (no of participants) is still low, possibly leading to misinterpretation due to inadequate information. Although I appreciate the huge efforts the authors made, this work needs more set of data to make a clearer conclusion, and therefore not suitable to be published.
Author Response
Thank you for the constructive peer review of our manuscript for Sensors. We have carefully considered each comment and applied related changes to the manuscript where appropriate. In our attached response, we address each comments and suggestions on a point-by-point basis (with the reviewer’s original comments highlighted in blue for clarity), and quote sections of the manuscript that have been changed as a result. In the new version of the manuscript that we will upload all changes have been tracked using the ‘track-changes’ function in Word.

Reviewer 2 Report
Congratulations on your study. Excellent presentation. I do feel that with such a small sample size, you need more evidence to support the significance of your study. With such a small sample size, it is actually difficult to make conclusions. Also, parametric testing is almost impossible to be done. Furthermore, using SD in a study with 8 participants means absolutely nothing. I would prefer writing this paper as a technical report rather than a clinical trial. Make sure that you include a power analysis for future studies. Your methods are very good.
Author Response

(The authors gave the same response as above.)

Reviewer 3 Report
The research is very interesting, sufficiently comprehensively described, and very well logically organized (I mean not only the design of the experiment but the paper description, as well). More details about the wearable are welcome (if possible structure and materials involved, the way of signals measurement, transduced and transmitted for collection). The manuscript is strongly recommended for acceptance and publication after adding this small piece of information!
Author Response
Thank you for the constructive peer review of our manuscript for Sensors. We have considered your comment and applied related changes to the manuscript where appropriate. In our attached response, we address the comments and suggestions on a point-by-point basis (with the reviewer’s original comments highlighted in blue for clarity), and quote sections of the manuscript that have been changed as a result. In the new version of the manuscript that we will upload all changes have been tracked using the ‘track-changes’ function in Word.

Round 2
Reviewer 1 Report
Thank you for your clarification. The author has addressed and modified the manuscript as concerned.
Reviewer 2 Report
Congratulations! I think that the manuscript has been improved enough for publication.